# WHERE CAN QUANTUM KERNEL BASED CLASSIFICATION METHODS MAKE A BIG DIFFERENCE?

## ABSTRACT

The classification problem is a core problem of supervised learning, which is widely present in our life. As a class of algorithms for pattern analysis, Kernel methods have been widely and effectively applied to classification problems. However, when very complex patterns are encountered, the existing kernel methods are powerless. Recent studies have shown that quantum kernel methods can effectively handle some classification problems of complex patterns that classical kernel methods cannot handle. However, this does not mean that quantum kernel methods are better than classical kernel methods in all cases. It is still unclear under what circumstances quantum kernel methods can realize their great potential. In this paper, by exploring and summarizing the essential differences between quantum kernel functions and classical kernel functions, we propose a criterion based on inter-class and intra-class distance and geometric properties to determine under what circumstances quantum kernel methods will be superior. We validate our method with toy examples and multiple real datasets from Qiskit and Kaggle. The experiments show that our method can be used as a valid determination method.

## 1 INTRODUCTION

Since the birth of quantum computing, researchers have been looking for the best place to apply quantum algorithms. The first two quantum algorithms devised were Grover (1996) and Shor (1994). They proved the advantages of quantum algorithms in some specific problems such as search and factorization theoretically. With the rise of artificial intelligence, more and more quantum algorithms combine quantum computing with machine learning. For example, quantum neural network (QNN) was first proposed by Ezhov & Ventura (2000) but was only defined in general terms at the physical level. Ricks & Ventura (2003) defined an approach to train QNNs, but the complexity of its method is exponential. Subsequently, Lloyd et al. (2013), Blacoe et al. (2013), and Rebentrost et al. (2014) tried to introduce quantum computing into clustering, distributed semantics, and SVMs, respectively, but their approaches were too limited to theory. As researchers introduce quantum into various machine learning algorithms, Schuld et al. (2015), Biamonte et al. (2017), Kopczyk (2018), Ciliberto et al. (2018) have started to summarize and sort out the concept of quantum machine learning.

The physical implementation of quantum computers has made great strides in recent years. In 2019, Arute et al. (2019) announced the achievement of quantum hegemony, a milestone event in quantum computing. Also, thanks to the emergence of several quantum computing platforms, such as those of IBM and Google, it has become possible for ordinary researchers to translate their research on quantum machine learning algorithms from theory to practice. For example, Farhi & Neven (2018), Quek et al. (2021), Verdon et al. (2019), Garg et al. (2019), Srivastava et al. (2020), and Meichanetzidis et al. (2020) have demonstrated the value of quantum machine learning in machine learning tasks by practice, respectively.

On the other hand, one of the most famous machine learning algorithms is the kernel method. A detailed description of the kernel method has been given by Burges (1998a), Muller et al. (2001), Scholkopf (2001) and Hofmann et al. (2008). Inspired by the classical kernel approach, Rebentrost et al. (2014) proposed a quantum kernel approach based on SVM, but only theoretically feasible.

It was not until Schuld & Killoran (2019), and Havlíček et al. (2019) systematically proposed two feasible implementations of quantum kernel methods that made quantum kernel methods became

one of the most mature and practically valuable quantum machine learning methods. Later, Blank et al. (2020), Wang et al. (2021), Kusumoto et al. (2021), and Peters et al. (2021) experimentally demonstrated the superiority of quantum kernel methods on some datasets. However, none of them have systematically explored the conditions when quantum kernel methods exist to their advantage. Schuld (2021) summarizes the connection between quantum kernel methods and classical kernel methods. However, it is still unclear when quantum kernel methods will have advantages over classical kernel methods. In this paper, we conclude under what circumstances the quantum kernel method is better or worse than the classical kernel method. Specifically,

- We propose that the quantum kernel function is probabilistic and classify the existing kernel functions.

- We propose a distance-based criterion $\delta$ to determine whether the quantum kernel method has the quantum advantage for a given dataset and demonstrate this experimentally.

- We explore and find the relationship between the superiority of quantum kernel methods in two dimensions and (1) the complexity of the data pattern (2) the data based on Mersenne Twister random distribution.

## 2 BACKGROUND AND RELATED WORK

**Classical Kernel.** Kernel methods, summarised by Muller et al. (2001) and Hofmann et al. (2008), are an important class of machine learning methods that carry out machine learning by defining which data points are similar to each other and which are not. Mathematically, the similarity is a distance in the data space, i.e., the distance between digital representations of data points. Specifically, the kernel method uses a feature mapping function $f_c$ to map data from a point in the original input space $\mathcal{O}$ to a higher-dimensional Hilbert feature space $\mathcal{F}_c$, i.e., $f_c : \mathcal{O} \to \mathcal{F}_c$, making separability between data classes more explicit. One of the most famous methods is the support vector machine (SVM) proposed by Burges (1998b).

One important factor that makes the kernel method successful is the kernel track. Scholkopf (2001) pointed out that instead of explicitly calculating the distance in high-dimensional Hilbert space, this distance can be calculated implicitly in low-dimensional input space by the kernel function $K$, but with the same effect. It can reduce the computational effort significantly and avoid a large number of calculations. A nonlinear classification problem is one of the classical machine learning problems, and kernel methods can effectively handle such problems. Recall that in classical kernel methods, such as support vector machine, a data point $x_i \in \mathbb{R}^n$ is mapped into a potentially much higher dimensional feature space $\mathcal{F}_c$ via a nonlinear mapping function $f_c$, where $x_i$ is represented by $\phi(x_i)$, i.e., $f_c : x_i \to \phi(x_i)$. In space $\mathcal{F}_c$, the nonlinear classification problem becomes a linear classification problem and simplifies the problem. The inner product of $\phi(x_i)^T \phi(x_j)$ is often seen as distances between $x_i$ and $x_j$ in the new space $\mathcal{F}_c$.

**Quantum Kernel.** The quantum kernel method is a kernel method designed to run on quantum computers based on quantum computing properties. Its principle is almost identical to the classical kernel method except that it maps the data point from the original input space $\mathcal{O}$ to the quantum Hilbert space $\mathcal{F}_q$, i.e., $f_q : \mathcal{O} \to \mathcal{F}_q$. The key to the quantum kernel methods is the quantum mapping function $f_q$. We can view the feature mapping function $f_q$ as the key to define the quantum kernel methods. Thus, if the quantum kernel approach is superior, the superiority lies in the quantum mapping function. The mechanism of quantum kernel methods is basically the same as that of classical kernel methods. The data point $x_i$ is mapped from the original input space $\mathcal{O}$ to the quantum state space $\mathcal{F}_q$, i.e., $f_q : x_i \to |\phi(x_i)\rangle$, where the $|\cdot\rangle$ denotes a vector and physically it represents a state of some quantum system. The $\langle\cdot|$ is the *Hermitian Conjugate* of the vector $|\cdot\rangle$. In practice, the feature map is realized by acting the circuit $U(x_i)$ on the initial quantum state $|0^n\rangle$, i.e.,

$$|\phi(x_i)\rangle = U(x_i) |0^n\rangle . \tag{1}$$

The quantum kernel can be obtained by running the circuit $U^\dagger(x_j)U(x_i)$ on the initial quantum state $|0^n\rangle$, where $U^\dagger$ is the *Hermitian conjugate* of $U$. Then estimate $|\langle 0^n| U^\dagger(x_j)U(x_i) |0^n\rangle|^2$ by counting the frequency of the $0^n$ output as a value of $k(x_i, x_j)$. Fig.1A shows the process flow of the quantum kernel method and classical kernel method.

**Quantum Kernel Method Based On Pauli Feature Map.** Following the IBM quantum computing platform, we take two qubits as an example. The general expression of a 2-qubit quantum kernel is

$$k(x_i, x_j) = |\langle \phi(x_i) | \phi(x_j) \rangle|^2 = |\langle 0^2 | U^\dagger(x_j) U(x_i) | 0^2 \rangle|^2. \tag{2}$$

By the definition of Havlíček et al. (2019), the quantum circuit $U$ is realized by $U(\vec{x}) = U_{\phi(\vec{x})} H^{\otimes 2} U_{\phi(\vec{x})} H^{\otimes 2}$, where the $\otimes$ is the *Kronecker Product* of two matrices. For the Second-order Pauli-Z evolution circuit, $U_{\phi(\vec{x})} = exp(i(x_0 Z_0 + x_1 Z_1 + (\pi - x_0)(\pi - x_1) Z_0 Z_1))$, where $Z_0$, $Z_1$ are quantum *Z-Gate*s, and $H$ is the quantum *Hadamard-Gate*. We denote the corresponding quantum kernel method as the *Z-ZZ quantum kernel method*, and the corresponding feature map is showed in Fig.1B(3). The feature maps of the *Z quantum kernel method* and the *ZZ quantum kernel method* are shown in Fig.1B(1) and Fig.1B(2), respectively. In this paper, all references to quantum kernel methods refer to the *Z-ZZ quantum kernel method* unless otherwise stated. In section 4.3, we compare these three quantum kernel methods.

**Support Vector Machine.** Support Vector Machine is a maximal margin classifier. It is seen as one of the most successful cases of the kernel approach. SVMs are dedicated to finding a hyper-plane that separates different classes and makes the margin as large as possible. In general cases, i.e., nonlinear cases, the data is mapped non-linearly to high dimensional Hilbert space by a mapping function. Then the distance between two data points can be calculated using the kernel function. Suppose we have a set of data points $D = \{(x_1, y_1), \cdots, (x_n, y_n)\}$, where $x_i \in \mathbb{R}^d$ and $y_i \in \{-1, +1\}$. According to Burges (1998b), the nonlinear SVM can be modified and expressed by an optimization problem as maximize:

$$L_D \equiv \sum_i^n \alpha_i - \frac{1}{2} \sum_{i,j=0}^n \alpha_i \alpha_j y_i y_j k(x_i, x_j) \tag{3}$$

s.t. $0 \le \alpha_i \le C$ and $\sum_i^n \alpha_i y_i = 0$, where $i = 1, \cdots, n$. The decision function is

$$f(x) = sign(\sum_{i=1}^{N_s} \alpha_i y_i \phi(s_i)^T \phi(x) + b) = sign(\sum_{i=1}^{N_s} \alpha_i y_i k(s_i, x) + b), \tag{4}$$

where $s_i$ are the support vectors and $N_s$ is the number of support vectors. The SVM-based quantum kernel method is very similar in principle to the traditional SVM, except that the computation of the kernel is performed on a quantum computer. Havlíček et al. (2019) refer to it as quantum kernel estimation. We show the specific estimation method in Section 4.2.

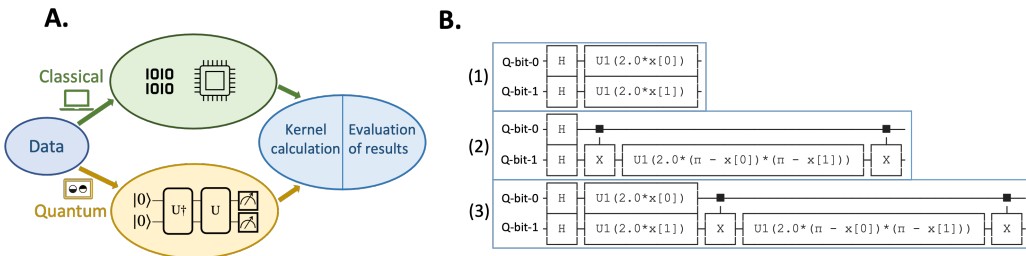

Figure 1: A. Working process of classical kernel method and quantum kernel method. B. Based on the IBM quantum computing platform, (1), (2), and (3) denote the feature maps of *Z*, *ZZ*, and *Z-ZZ* quantum kernel methods, respectively. Note that we only show one repetition here, and in the experiments, the number of repetitions of each method is set to two.

## 3 METHODS

### 3.1 QUANTUM KERNEL IS A PROBABILISTIC KERNEL

The kernel function is an equation for measuring similarity. In vector space, we estimate the similarity of vectors utilizing vector kernel functions. Similarly, graph kernels describe the similarity of two graphs, and tree kernels compare the similarity of trees, which are often used in natural language

processing. A question arises as to how to define kernels or what kind of kernel functions can be effectively used or not. There is no answer to this question. Mercer (1909) argues that a valid kernel function needs to satisfy symmetry and positive definiteness. However, some kernel functions that do not obey Mercer's condition still achieve good results in some specific tasks, such as the widely used sigmoid kernel function proposed by Lin & Lin (2003). Mix kernel function proposed by Smits & Jordaan (2002) tries to achieve better properties when combining different kernel functions.

The mechanism of the quantum kernel function is similar to some traditional kernel functions. It follows the Mercer theorem and is a practical kernel function that expands the family of kernel functions. However, its implementation is based on quantum superposition states and entanglement. Since the values obtained are based on probabilities in a statistical sense, we call it a probabilistic kernel function. For correspondence, we call the classical kernel function a deterministic kernel function. We try to clearly show the relationship and difference between different kernel functions by a diagram (Fig.2). It is worth noting that there are no guarantees for one kernel to work better than the other in all cases, according to the *No Free Lunch Theorem* (Wolpert & Macready (1997)). Choosing different kernel functions in various subjects will achieve better results. The primary purpose of this paper is to investigate under what circumstances the quantum kernel method is better or worse than the classical kernel method.

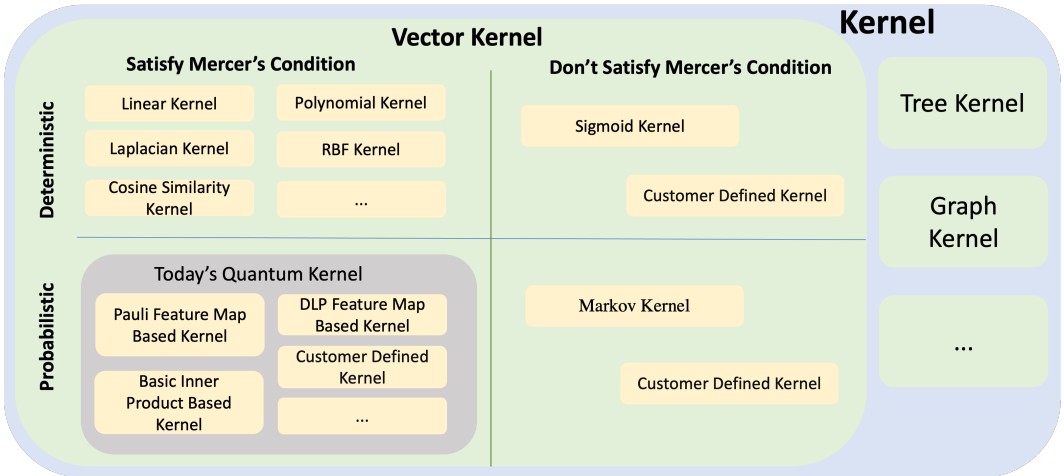

Figure 2: Kernels Category

### 3.2 THE PROPOSED PATTERNS AND CRITERIA FOR JUDGING QKM

First, we would like to demonstrate the advantages of quantum kernel methods over classical kernel methods when dealing with data patterns based on the Mersenne Twister random distribution. The Mersenne Twister is a pseudo random number generator which was first proposed by Matsumoto & Nishimura (1998). The Mersenne Twister is used as default pseudo random number generator by many software, such as Python, R, and PHP. The Mersenne Twister random distribution is a distribution that be generated by the Mersenne Twister method.

**Theorem 3.1 (Advantages of QKM for random distributions)** *In two dimensions, the Z-ZZ quantum kernel method has better learning ability than classical kernel methods for randomly distributed data patterns based on Mersenne Twister.*

*Proof.* We assume that the *Z-ZZ* feature map can effectively simulate the efficacy of the feature map proposed by Liu et al. (2021). By Matsumoto & Nishimura (1998), for a k-bits binary number, the Mersenne Twister algorithm generates discrete uniformly distributed random numbers in the range $[0, 2^k - 1]$. Solving this problem is a discrete logarithm problem (DLP). For DLP, Liu et al. (2021) say no efficient classical algorithm can achieve inverse-polynomially better accuracy than random guessing. Therefore, the *Z-ZZ* quantum kernel method can demonstrate quantum superiority over the classical kernel methods for Mersenne Twister-based randomly distributed data patterns.

Second, we demonstrate that whether quantum kernel methods have quantum advantages is related to the $\delta$ of the data set, where $\delta$ will be defined in Equ.(7). Because any classification problem can be transformed into a binary situation, we all base our study on the binary classification problem. If the distance between classes is large enough, i.e., the simple case, the quantum kernel methods are not as good as the classical kernel methods. Based on fourteen Adhoc-Modify datasets (Section 4.1 Qiskit package datasets (6)), we measure the $\delta$ of each dataset and draw a graph with $\delta$ as the horizontal coordinate. Fig.3 shows that when $\delta$ increases to a certain level, the advantage of the quantum kernel method disappears.

To measure the degree of separation of two classes $C_l$ and $C_m$, we first define the inter-class distance

$$D(C_l, C_m) = \frac{1}{N_l N_m} \sum_{k=1}^{N_l} \sum_{j=1}^{N_m} d(x_k^{(l)}, x_j^{(m)}), \qquad (5)$$

where $N_l$ and $N_m$ are the number of observations belonging to class $C_l$ and $C_m$, respectively. The $x_n^{(l)}$ is the n-th sample in class $C_l$ and $d(,)$ is the euclidean distance between two samples, i.e., $d(\vec{x}, \vec{y}) = \sqrt{\sum_{i=1}^{n}(x_i - y_i)^2}$.

For the sake of uniformity, we also need to define the intra-class distance for class C, i.e.,

$$D(C) = \frac{1}{N(N-1)} \sum_{k=1}^{N} \sum_{j=1, j \neq k}^{N} d(x_k, x_j), \qquad (6)$$

where the N is the of class $C$, and $x_n$ is the n-th sample in class $C$. We propose a criterion to evaluate whether quantum kernel methods will be better than classical kernel methods. In a binary classification problem, we define the degree of integration of $C_l$ and $C_m$ as $\delta_{lm}$.

$$\delta_{lm} = \frac{D(C_l, C_m)}{D(C_l) + D(C_m)} \qquad (7)$$

Theorem 3.2 illustrates that the larger the $\delta$, the greater the separation of the two classes. When the delta is large enough, the quantum kernel function loses its quantum advantage.

**Theorem 3.2 (Deficiencies of the QKM)** *In the case of a balanced number of the two classes, there exists $\delta_0$ such that the quantum kernel method will not be better than the classical kernel method in handling classification problems when $\delta > \delta_0$. In practice, $\delta_0$ is usually taken as 0.6.*

*Proof.* Suppose our measurement independent identical distribution $M_1, M_2 \cdots M_R$ which have expectation $E(M) = \mu$ and variance $D(M) = \sigma^2$, where M is the random variable, R is the number of measurement shots. By the Central Limit Theorem, for any $m$, the distribution function $F_R(m) = P\{\frac{\sum_{i=1}^{R} M_i - RE}{\sqrt{DR}} \leq m\}$ satisfies: $\lim_{R \to \infty} F_R(x) = \lim_{R \to \infty} \{\frac{\sum_{i=1}^{R} M_i - RE}{\sqrt{RD}} \leq x\} = \frac{1}{\sqrt{2\pi}} \int_{-\infty}^{m} e^{-\frac{t^2}{2}} dt$. This shows that when R is large enough, the random variable $Y_R = \frac{\sum_{i=1}^{R} M_i - RE}{\sqrt{RD}}$ obeys normal distribution $N(0, 1)$. So, $\sum_{i=1}^{R} M_i = \sqrt{RD} Y_R + RE = \sqrt{R} \sigma Y_R + R\mu$ obeys normal distribution $N(R\mu, R\sigma^2)$. However, even fully error corrected, the quantum methods still have the noise caused by measurement. Two types of data are linearly separable or almost linearly separable when $D(C_l, C_m) \gg D(C_l) + D(C_m)$, i.e., $\delta \gg \delta_0$. It is easy for a deterministic kernel method to find a boundary line with an infinitely small error. So, when $\delta > \delta_0$, the quantum kernel method is not better than the classical kernel method in handling classification problems.

In fact, as a probabilistic kernel method, a quantum kernel method has more error than a deterministic kernel method caused by the measurement process. It is especially evident in some simple cases because we assume that deterministic kernel methods must find a boundary line. However, errors in probabilistic methods are inevitable.

## 4 EXPERIMENTS

### 4.1 DATA PREPARATION

**Qiskit package datasets.** Five datasets from qiskit.ml.datasets were used in the experiments. (1) Digits datasets. This dataset consists of 1797 8×8 images, and each image is a handwritten number

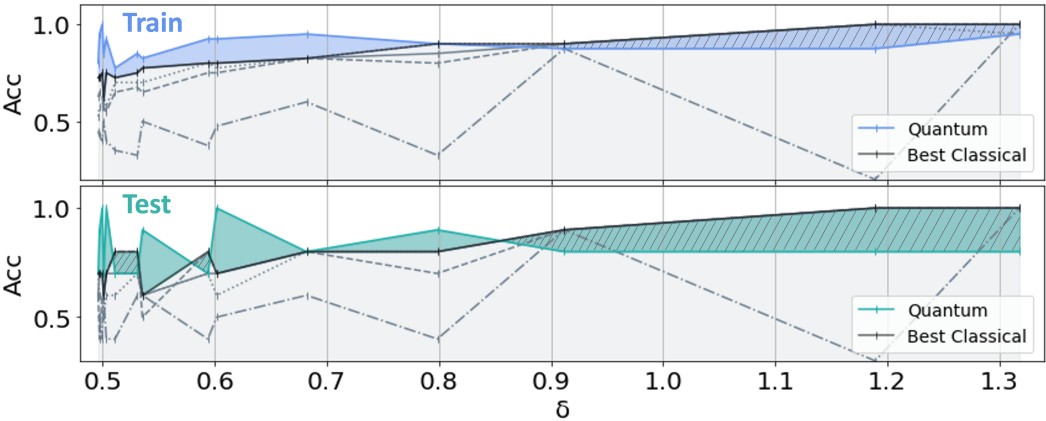

Figure 3: The relationship between $\delta$ and the predicting accuracy in Ad hoc dataset. The top graph shows the training accuracy, and the bottom graph shows the testing accuracy.

0~9. We perform binary classification for every two digits. Thus, 45 small datasets were generated. (2) Breast cancer dataset. (3) Iris dataset. (4) Wine dataset. (5) Adhoc dataset. Note that (1)~(4) are copies from UCI ML Hand-written Digits Dataset, UCI ML Breast Cancer Wisconsin (Diagnostic) Dataset, UCI ML Iris Plants Dataset, and UCI ML Wine Recognition Dataset, respectively. For (5) Adhoc dataset, we separate the two classes and translate all the data points in one class by the same distance, leaving the other class unchanged. In this way, we get 14 datasets, denoted as (6) Adhoc-Modify datasets.

**Kaggle datasets.** Five datasets from Kaggle were used in the experiments. (7) Email spam dataset (Balaka Biswas), (8) Heart disease dataset (Zeeshan Mulla). (9) Giants and dwarfs dataset (Vinesmsuic). (10) Star type dataset (Baris Dincer). (11) Drug dataset (Pratham Tripathi).

**Geometric toy datasets.** We designed the two-dimensional geometric toy datasets to illustrate the learning ability of the quantum kernel method for processing classification problems in geometric patterns. (12) Geometric non-random datasets. We designed 2~5 layers of concentric circles and four squares with different mixing degrees. Fig.4A shows the four circular datasets and four square datasets. (13) Geometric random datasets. Based on the Mersenne Twister random distribution, we designed datasets with random distributions in three geometric patterns: Circular, square, and equilateral triangle patterns. The three graphs on the right side of Fig.4B show a sample of these three datasets, respectively.

## 4.2 PROCESS OF TRAINING AND TESTING OF QUANTUM KERNEL METHODS

We briefly introduce the training process of the quantum kernel method here, following the idea of Liu et al. (2021). Suppose a training dataset $D_{train} = \{(x_1, y_1), ..., (x_n, y_n)\}$. Since the only difference of the quantum kernel method in an SVM from the classical SVM is kernel calculation, here we only show the kernel calculation process in a quantum computer.

**Training:** For each pair of data points $x_i$ and $x_j$ ($i \neq j$) in $D_{train}$, we apply $U^\dagger(x_i)U(x_j)$ on the input $\left|0^{\otimes 2}\right\rangle$. After $R$ repetitive runs, we record the number of times the output results in $\left|0^{\otimes 2}\right\rangle$, denoted as $R_0$ and $k(x_i, x_j) = \frac{R_0}{2R}$, where $i \neq j$ and $k(x_i, x_i) = 1$. In the end, we apply Equ.(3) directly. **Testing:** When there comes a new sample $x_{new}$, for each data point $x_i$ in $D_{train}$, we apply $U^\dagger(x_i)U(x_{new})$ on the input $\left|0^{\otimes 2}\right\rangle$. After $R$ repetitive runs, we record the number of times the output results in $\left|0^{\otimes 2}\right\rangle$, denoted as $R_1$ and $k(x_{new}, x_i) = \frac{R_1 + 2R}{2R}$. In the end, we apply Equ.(4) directly.

## 4.3 EXPERIMENTS RESULTS

**The quantum kernel method is good at complex data patterns.** In this part, we explore the ability of quantum kernel methods to solve classification problems under challenging patterns. Firstly, we increase the learning difficulty by increasing the complexity of the geometric patterns and get four patterns, i.e., P1~P4 in Fig.4A. Experiments show that from P1 to P4, the classical kernel approach, which initially performs better, gradually loses its superiority. In contrast, the superiority of the quantum kernel approach begins to emerge.

To further increase the learning difficulty, we hypothesized the existence of a pattern in which the data are randomly distributed over the geometry according to the Mersenne Twister. The right three graphs in Fig.4B show this kind of data pattern. To explore the learning ability of quantum kernel methods for Mersenne Twister randomly distributed data patterns, we plotted the relationship between the size of the dataset and the prediction accuracy of each method. Considering the randomness of the random distribution, we take the average result of 50 times as the result of each experiment. The experiments demonstrate that the quantum kernel method has a more robust learning capability than the classical kernel method in this complex pattern.

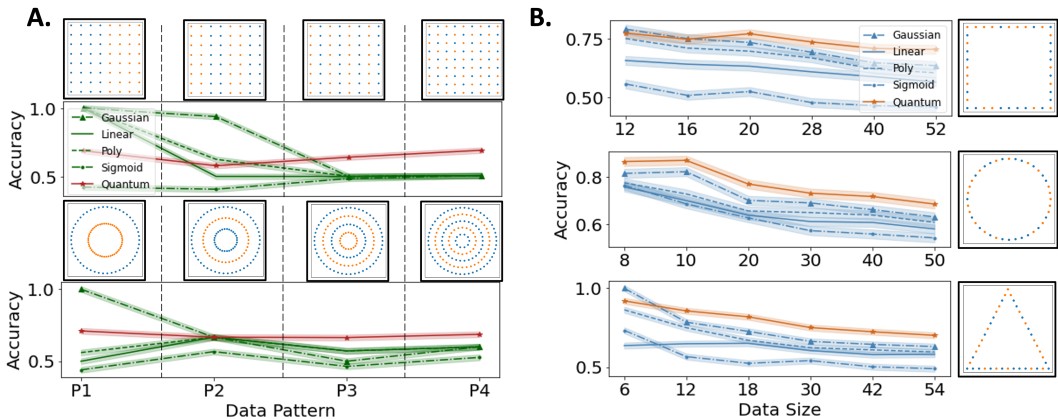

Figure 4: A. The horizontal coordinates P1~P4 represent the data patterns of different difficulties of the geometric non-random dataset, respectively. B. The horizontal coordinates indicate the dataset size, and the vertical coordinates indicate the prediction accuracy. B illustrates the relationship between the prediction correctness and the size of the dataset under three patterns in the geometric random distribution dataset.

**The quantum kernel method fails at large values of $\delta$.** In this part, we will show that our criteria can effectively determine whether quantum kernel methods can demonstrate advantages over classical kernel methods or not. First, taking the Digits dataset as an example, we perform binary classification for every two digits. Thus, we generate 45 small datasets. We then use PCA to reduce the dimensionality of all data to 2 dimensions, as shown in Fig.5. We recorded each small dataset's training and test accuracy after applying four classical kernel methods and a quantum kernel method. The bottom left panel in Fig.5 shows the prediction accuracy of each method. According to the experimental results, we attach a corner mark to each data set. The red rounded corner markers indicate that the quantum kernel method has the potential for quantum dominance on the corresponding dataset, while the black rounded corner markers indicate that it does not. The blue corner markers indicate that the quantum kernel method is indistinguishable from the classical kernel method. Intuitively, the quantum kernel approach for linearly divisible data sets does not show superiority. However, for complex models with relatively high fusion, the quantum kernel approach has the potential to be superior.

To validate our criteria $\delta$, we prepared 81 datasets: 45 small datasets from Digits, 14 datasets from Adhoc-Modify, Breast cancer, Iris, Wine, Email spam, Heart disease, Giants and dwarfs, Star Type, Drug, 4 datasets from Geometric non-random, i.e., the top-left, bottom-left, top-right and bottom-right four datasets in Fig.4A, 10 datasets from Geometric random including five circles of size thirty and five squares of size forty. The difference in prediction accuracy between classical kernel methods

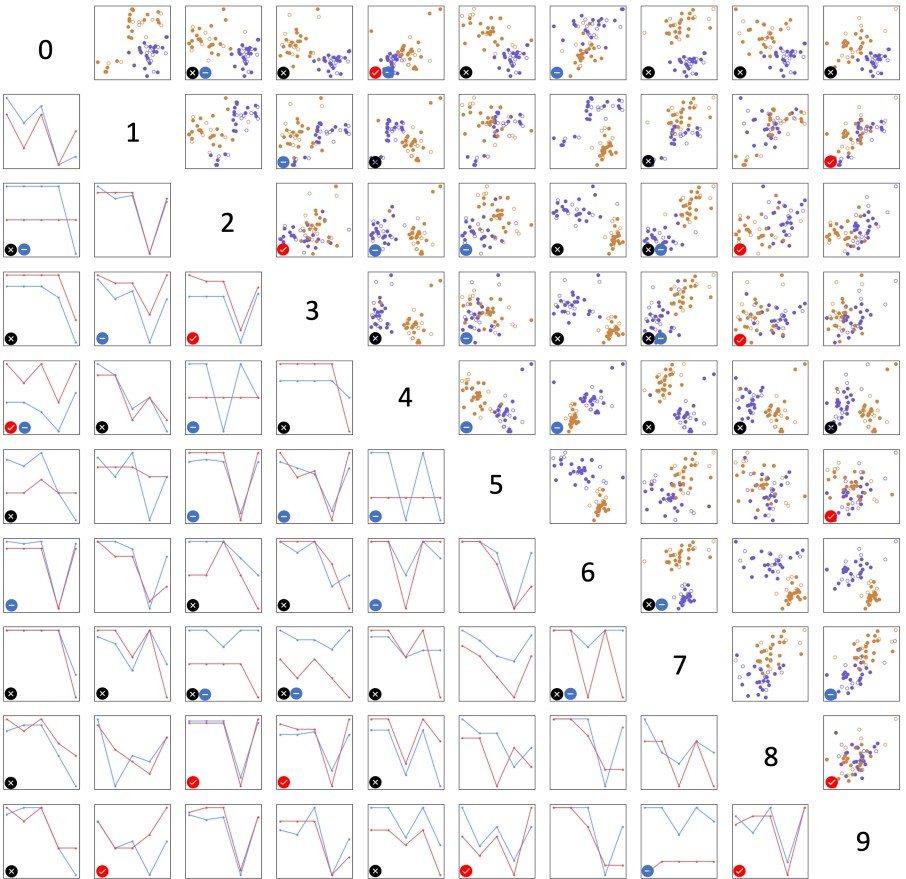

Figure 5: The top right panel shows the 45 datasets after visualization, and the bottom left panel shows the prediction accuracy of the five methods, which are SVM models based on Gaussian kernel, linear kernel, polynomial kernel, sigmoid kernel, and quantum kernel, respectively. The blue line represents the training set, and the red represents the test set. Red corners indicate the case where the training or test scores of the quantum kernel method outperform all classical kernel methods. Black corners indicate the case where the training or test scores of the quantum kernel method are worse than any classical kernel method. Blue corners indicate the case where the training or test scores of the quantum kernel method are equal to the best value of all classical kernel methods.

and quantum kernel methods on each dataset is calculated. It is shown by the relevant vertical lines in Fig.6. The magnitude of $\delta$ on each dataset is shown with an asterisk in Fig.6. According to the experiments, when the quantum kernel method has an advantage, the asterisks appear below the 0.6 level line all the time. We can get the conclusion that (i) the availability of quantum superiority correlates with $\delta$ and (ii) the quantum kernel method can be better than the classical kernel method when $\delta < 0.6$, although it is not a sufficient condition.

**Simple quantum kernel methods do not offer quantum advantages.** When the feature space is so large that the kernel function is computationally expensive, quantum kernel methods can effectively estimate their kernel functions, but classical kernel methods cannot. So, quantum kernel methods are preferred over classical kernel methods for classification problems with complex patterns. However, a simpler quantum kernel function can be simulated classically. For a simple-kernel-based quantum kernel method, it no more has the superiority. Its ability to handle complex pattern problems is significantly reduced and even inferior to classical kernel methods. Interestingly, sometimes this simple quantum kernel function works well with simple pattern problems.

We illustrate the above opinion experimentally. In this part, we prepared 68 datasets: 45 small datasets from Digits, Breast cancer, Iris, Wine, Adhoc, Email spam, Heart disease, Giants and dwarfs, Star Type, Drug, 4 datasets from Geometric non-random, i.e., the top-left, bottom-left, top-

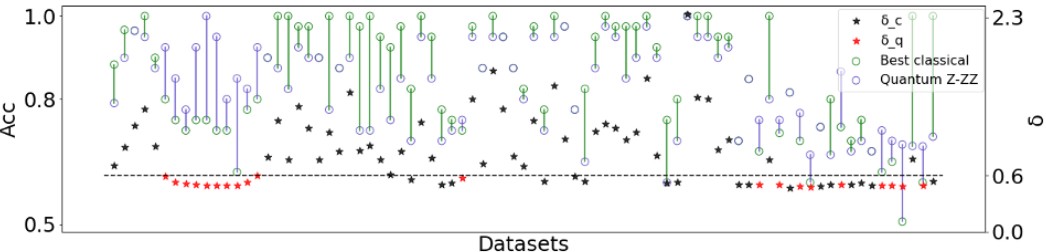

Figure 6: The horizontal coordinate is the 81 training datasets, the left vertical coordinates indicate the method prediction accuracy, and the right vertical coordinates indicate the value of $\delta$. The green circles represent the best classical kernel methods in RBF, Linear, Polynomial, and Sigmoid kernel-based kernel methods. The purple circles represent the quantum kernel methods. Under the same dataset, if the classical kernel method is not worse than the quantum kernel method, we use the green vertical line to indicate how better the classical kernel method is than the quantum kernel method and the black asterisk to indicate the magnitude of $\delta$. Otherwise, we use the purple vertical line to indicate how better the quantum kernel method is than the classical kernel method and use the red asterisk to indicate the magnitude of $\delta$.

right and bottom-right four datasets in Fig.4A, 10 datasets from Geometric random including five circles of size thirty and five squares of size forty. By comparing the performance of the best classical kernel method and the three quantum kernel methods on 68 training datasets, we find that some kernel methods based on simple quantum kernel functions, such as the $Z$ quantum kernel method, do outperform the $Z$-$ZZ$ kernel method on some datasets, but do not outperform the best classical kernel methods. However, when quantum superiority exists, it is often achieved by $Z$-$ZZ$ kernel methods. The experiments show that the quantum kernel method superiority on only one dataset is achieved by the $Z$ quantum kernel method.

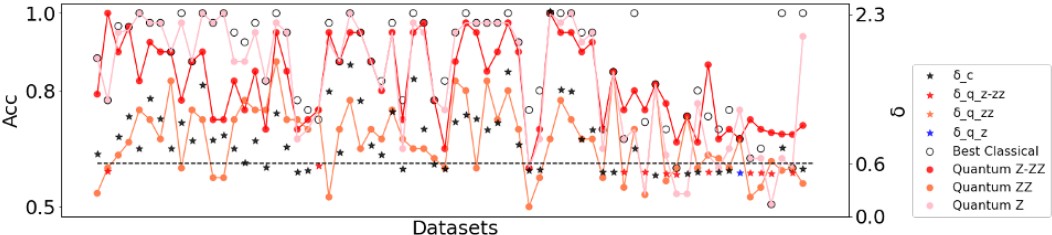

Figure 7: The horizontal coordinate is the 68 training datasets, the left vertical indicates the method's prediction accuracy, and the right vertical coordinate indicates the value of $\delta$. The hollow black circles, red circles, orange circles, and pink circles represent the best classical kernel methods, $Z$-$ZZ$ quantum kernel-based, $ZZ$ quantum kernel, and $Z$ quantum kernel-based quantum kernel methods, respectively. We use blue asterisks to indicate the value of $\delta$ when the $Z$-quantum kernel-based method is optimal, which has only one case, and orange asterisks to indicate the value of $\delta$ when the $ZZ$-quantum kernel-based method is the best, although this case does not occur in the experiment. Correspondingly, red and black asterisks are used to represent the value of $\delta$ when the $Z$-$ZZ$ quantum kernel method and the classical kernel method are best, respectively.

## 5 CONCLUSION

The classification problem is one of the most common problems in machine learning, and both classical kernel methods and quantum kernel methods can effectively handle the classification problem. However, it is difficult to determine what situation each of them is suitable for. This paper explores when quantum kernel methods can take quantum advantage by comparing different kernel functions. Moreover, a judgment criterion is proposed to help one decide when quantum kernel methods can achieve better results than classical kernel methods. Experiments show that our method is effective.

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
