# OpenReview forum: "Where can quantum kernel methods make a big difference?"
_ICLR.cc/2022/Conference — ICLR 2022 Submitted_

### Official Review · Reviewer_3XVL · 2021-11-02

**Correctness:** 2
**Technical Novelty And Significance:** 3
**Empirical Novelty And Significance:** 3
**Recommendation:** 5
**Confidence:** 4

**Main Review:**

The background section was clear and thorough.

I have some difficulties with section 3:

- proof of theorem 3.1.  The proof hinges on an assumption - "the Z-ZZ feature map can effectively simulate the efficacy of the feature map proposed by Liu et al. (2021)" - but I don't see any proof of this conjecture, at least in the mathematical sense.  If this is axiomatic to the theorem then it should be presented as such in the theorem: otherwise the proof doesn't work afaict.  Have I missed something here?

- theorem 3.2: why is $\delta_0$ usually taken as 0.6?  Is this based on the experimental results alone, or is there some intuition as to why this particular threshold is important?

However the experimental section is very thorough and quite convincing, so I am willing to overlook my misgivings regarding the theorems.

Minor point:

- in the paragraph before equation (1), is the definition of the feature map $f_q : x_i \to < \phi (x_i) | \phi  (x_j) >$ correct?  And if so, how does $x_j$ fit here?


-- EDIT AFTER FURTHER CONSIDERATION --

After further consideration and reading other comments I am persuaded that the informality of proofs is a more serious problem than I initially thought and adjusted my recommendation accordingly.

**Summary Of The Paper:**

The paper investigates the circumstances under which quantum kernel methods will be superior to classical kernel methods.  The criteria is based on a threshold of the ratio of the inter- and intra-class distances of the (binary) training data.  Validation of the proposed criteria is carried out on a range of toy and real datasets.

**Summary Of The Review:**

- Overall a well-written paper.
- I have some problems with the proof of theorem 3.1.
- I am also curious how the threshold value 0.6 is arrived at, and whether this is purely experimental or if there is some alternative intuition.
- The experimental section is very thorough and quite convincing.


-- EDIT AFTER FURTHER CONSIDERATION --

After further consideration and reading other comments I am persuaded that the informality of proofs is a more serious problem than I initially thought and adjusted my recommendation accordingly.

---

> ### Author Response · Authors · 2021-11-23
> **Response to reviewer4**
>
> Comment 1: I have some difficulties with section 3: proof of theorem 3.1. The proof hinges on an assumption - ”the Z-ZZ feature map can effectively simulate the efficacy of the feature map proposed by Liu et al. (2021)” - but I don’t see any proof of this conjecture, at least in the mathematical sense. If this is axiomatic to the theorem then it should be presented as such in the theorem: otherwise the proof doesn’t work afaict. Have I missed something here?
>
> Response 1: We appreciate the reviewer for recognizing our work. Some re- searchers such as [10] already proved that the quantum kernels can be superior to classical kernels when learning a DLP problem. In our paper, we know learning a Mersenne Twister distribution is a DLP problem. That’s why we assume that the Z-ZZ feature map can effectively simulate the efficacy of the feature map proposed by [10] in the beginning. Our experiments showed that the quantum kernel methods are superior to classical kernel methods. However, we cannot provide any proof of this conjecture at present. We will continue to focus and work on this in the future.
>
> Comment 2: theorem 3.2: Why is δ0 usually taken as 0.6? Is this based on the experimental results alone, or is there some intuition as to why this particular threshold is important?
>
> Response 2: We appreciate the time and efforts to review our work. We want to make clear the threshold 0.6 in Theorem 3.2. In our paper, we try to provide a threshold to decide which one is better to use a quantum kernel method or a classical kernel method. This threshold δ0 is a empirical quantity that is determined through several datasets (81 datasets in our experiment). The ”true” value of δ0 is unknown since it is a value determined by as many datasets as possible. We cannot try all the datasets to make sure of this. But, the existence of δ still makes sense. At least, we know a phenomenon that the variable δ can have some influence to decide whether a quantum kernel method is better or not when compared with a classical kernel method. Based on our experience, the δ0 will take 0.6. To illustrate how it works, we give an example here. Suppose we get the δ of a specific dataset D, if the δ for the D is larger than δ0, we have enough reason to believe that we can use classical kernels to learn this dataset. On the other hand, suppose the δ for the D is less or equal to δ0. According to theorem 3.2, even though we cannot directly say that the quantum kernels will be better, it at least provides us a choice to use quantum kernels. Whether the quantum kernels will be better depends on the data pattern. For example, as we mentioned in the paper, if we meet the Mersenne Twister random distribution, the quantum kernels will be superior.
>
> Comment 3: However the experimental section is very thorough and quite convincing, so I am willing to overlook my misgivings regarding the theorems. Minor point: In the paragraph before equation (1), is the definition of the feature map fq : xi → ⟨φ(xi)|φ(xj)⟩ correct? And if so, how does xj fit here?
>
> Response 3: Thanks for recognizing our work. We will continue to im- prove our work. There is a typo in the fq. We modified it as follow fq : xi → |φ(xi)⟩.

---

### Official Review · Reviewer_cR4C · 2021-11-02

**Correctness:** 1
**Technical Novelty And Significance:** 1
**Empirical Novelty And Significance:** 2
**Recommendation:** 1
**Confidence:** 4

**Main Review:**

1. The writing quality of the paper is not ideal. Sections 2 and 3 are long but not informative. The figures in experiments are hard to understand. The paper has a lot of grammar errors.

2. The paper seems to focus on the classification problems. Quantum kernels can be applied to a much wider range of problems. The terminology ``quantum kernel methods" in the title may need to be changed to quantum kernel based classification methods.

3. The proof of theorem 3.1 is just a summary of statements in the literature. The proof of theorem 3.2 is just an application of CLT (and with some errors). For example, the first sentence in the proof of Theorem 3.2. ``Suppose our measurement independent identical distribution ... where M is the random variable, R is the
number of measurement shots." is not correct in both English and mathematics. Also, I am not sure if the proof really did the job to prove the statements in Theorem 3.2.

4. The experiments are not convincing. Many implementation details are missing. The results, figures, and explanations are hard to understand. No replication codes are provided. My guess is the experiments are done by simulations run on the classical computer rather than real quantum computers?

**Summary Of The Paper:**

This paper studies quantum kernel methods for classification problems. It compares the differences between quantum kernel functions and classical kernel functions. A degree of integration and two theorems are proposed. Some empirical experiments are done on the datasets from Qiskit and Kaggle.

**Summary Of The Review:**

Correctness:
This paper only considered limited scenarios in classification problems. Therefore, the title and the statements in the paper may be exaggerated by saying quantum kernel methods instead of quantum kernel based classification methods. The proofs of the major theorems are not convincing to me. The experiments are not strong enough to support the claims made in the paper.

Technical Novelty And Significance:
The theoretical contributions of the paper are limited if there are some at all.

Empirical Novelty And Significance:
I am not sure if the authors have proposed any new quantum methods for the experiments. The results are hard to understand.

---

> ### Author Response · Authors · 2021-11-23
> **Response to reviewer3**
>
> Comment 1: The writing quality of the paper is not ideal. Sections 2 and 3 are long but not informative. The figures in experiments are hard to understand. The paper has a lot of grammar errors.
>
> Response 1: Thanks for the time and efforts to review our work. We will reconsider the whole paper carefully and continue to improve our work. We use much space to introduce the related work in section 2 since it involves much knowledge. In section 3 we try to introduce our contributions in detail. Fig.4, Fig.5, Fig.6, and Fig.7 are mainly about the comparison of quantum kernel methods and classical kernel methods based on different datasets. The high-level variable is the dataset. We will continue to make our work more readable and correct the grammar errors.
>
> Comment 2: The paper seems to focus on the classification problems. Quan- tum kernels can be applied to a much wider range of problems. The terminology “quantum kernel methods” in the title may need to be changed to quantum kernel-based classification methods.
>
> Response 2: Thanks for the helpful suggestion and we will modify the title. The title is going to be: ”Where Can Quantum Kernel-Based Classification Methods Make A Big Difference?”.
>
> Comment 3: The proof of theorem 3.1 is just a summary of statements in the literature. The proof of theorem 3.2 is just an application of CLT (and with some errors). For example, the first sentence in the proof of Theorem 3.2. “Suppose our measurement independent identical distribution ... where M is the random variable, R is the number of measurement shots.” is not correct in both English and mathematics. Also, I am not sure if the proof really did the job to prove the statements in Theorem 3.2.
>
> Response 3: Thanks for the advice. We will reconsider and continue to improve our theorems. Here we just want to make our theorems clearer.
> (1). Theorem 3.1 shows that quantum kernels are superior to classical kernels when meeting a random distribution based on Mersenne Twister Generator. To show this point, we designed an experiment, and the results are shown in Fig.4(B). The results show that the quantum kernels are almost always better than the classical kernels. As the number of data increases, the quantum kernels will maintain a stable advantage over classical kernels. We think it is an interesting phenomenon. Some researchers already proved that the quantum kernels can be superior to classical kernels when learning a DLP problem. Also, we know learning a Mersenne Twister distribution is a DLP problem. That’s why we assume that the Z-ZZ feature map can effectively simulate the efficacy of the feature map proposed by [10] in the beginning. However, we cannot provide rigorous mathematical proof at present. We will continue to focus and work on this in the future.
> (2) In theorem 3.2, we try to provide a threshold to decide which one is better to use a quantum kernel method or a classical kernel method. This threshold δ0 is an empirical quantity that is determined through several datasets (81 datasets in our experiment). The ”true” value of δ0 is unknown since it is a value determined by as many datasets as possible. We cannot try all the datasets to make sure of this. But, the existence of δ still makes sense. At least, we know a phenomenon that the variable δ can have some influence to decide whether a quantum kernel method is better or not when compared with a classical kernel method. Based on our experience, the δ0 will take 0.6. To illustrate how it works, we give an example here. Suppose we get the δ of a specific dataset D, if the δ for the D is larger than δ0, we have enough reason to believe that we can use classical kernels to learn this dataset. On the other hand, suppose the δ for the D is less or equal to δ0. According to theorem 3.2, even though we cannot directly say that the quantum kernels will be better, it at least provides us a choice to use quantum kernels. Whether the quantum kernels will be better depends on the data pattern. For example, as we mentioned in the paper, if we meet the Mersenne Twister random distribution, the quantum kernels will be superior.
>
> Comment 4: The experiments are not convincing. Many implementation details are missing. The results, figures, and explanations are hard to understand. No replication codes are provided. My guess is the experiments are done by simulations run on the classical computer rather than real quantum computers?
>
> Response 4: Thanks for the time and work to review our work. We will continue to improve our work. We can also apply the replication codes. The experiments are run on our local computer, but with the help of the IBM quantum platform where the bottom layer is a quantum computer.

---

### Official Review · Reviewer_DX2D · 2021-11-03

**Correctness:** 2
**Technical Novelty And Significance:** 2
**Empirical Novelty And Significance:** 4
**Recommendation:** 3
**Confidence:** 2

**Main Review:**

The introduction of the article is well-written and reflects a good knowledge of the literature. However, the authors restrict the comparison between quantum kernel methods and classical kernel methods to the case of binary classification. Moreover, the theoretical results are presented informally.

**Summary Of The Paper:**

This article studies the superiority of quantum kernel methods over classical kernel methods. The authors relate the quantum advantages for classification tasks to the separation between the classes.  Several numerical simulations were conducted to support the theoretical findings.


**Summary Of The Review:**

Unfortunately, I have to recommend that the paper would be rejected for the following reasons:

1.  the analysis concerns binary classification, which is one among many learning tasks to study. It is not clear how to deal with other learning tasks.
2. The presentation of the theoretical results (Theorem 3.1 and Theorem 3.2) is poor and informal, and the proofs lack rigor. For example, the threshold 0.6 in Theorem 3.2 Is mysterious and does not seem to stem from any grounded argument.

---

> ### Author Response · Authors · 2021-11-23
> **Response to reviewer DX2D**
>
> Comment 1: The introduction of the article is well-written and reflects a good knowledge of the literature. However, the authors restrict the comparison between quantum kernel methods and classical kernel methods to the case of binary classification. Moreover, the theoretical results are presented informally.
>
> Response 1: Thanks for the time and efforts to review our work. Yes, we only restrict the comparison to the case of binary classification. The binary classification problem is the basic classification problem. Any multi-class classification problem can be divided into binary classification problems. There are two methods to transfer a multi-class classification problem into a binary classification problem, i.e., ”one vs one” and ”one vs rest”. We will continue to do expand the content of our paper. What’s more, the theoretical results are almost based on our experiments. The threshold δ0 is an empirical value based on our experiments. We will try our best to make the results more formal.
>
> Comment 2: The analysis concerns binary classification, which is one among many learning tasks to study. It is not clear how to deal with other learning tasks.
>
> Response 2: Thanks for your constructive comments. In this paper, we only focus on a very common problem, the binary classification problem. For machine learning tasks, the two most popular tasks are classification and regression. Classification tasks include binary classification problems and multi-class classification problems. Multi-class classification problems can be transferred to binary classification problems using ”one vs one” or ”one vs rest” methods. As far as we know, there exist some quantum kernel methods that can deal with regression tasks, with a similar quantum mechanism. In our paper, we would like to reveal some laws to evaluate quantum kernel methods and classical kernel methods. So, we just start from a basic situation (the binary classification problem). We know there is still a lot of work to do and we will continue to work on other learning tasks.
>
> Comment 3: The presentation of the theoretical results (Theorem 3.1 and Theorem 3.2) is poor and informal, and the proofs lack rigor. For example, the threshold 0.6 in Theorem 3.2 is mysterious and does not seem to stem from any grounded argument.
>
> Response 3: Thanks for your comments. We will think about them carefully and reconsider the presentation. We also want to make clear the threshold 0.6 in Theorem 3.2. In our paper, we try to provide a threshold to decide which one is better to use a quantum kernel method or a classical kernel method. This threshold δ0 is an empirical quantity that is determined through several datasets (81 datasets in our experiment). The ”true” value of δ0 is unknown since it is a value determined by as many datasets as pos- sible. We cannot try all the datasets to make sure of this. But, the existence of δ still makes sense. At least, we know a phenomenon that the variable δ can have some influence to decide whether a quantum kernel method is better or not when compared with a classical kernel method. Based on our experience, the δ0 will take 0.6. To illustrate how it works, we give an example here. Suppose we get the δ of a specific dataset D, if the δ for the D is larger than δ0, we have enough reason to believe that we can use classical kernels to learn this dataset. On the other hand, suppose the δ for the D is less or equal to δ0. According to theorem 3.2, even though we cannot directly say that the quantum kernels will be better, it at least provides us a choice to use quantum kernels. Whether the quantum kernels will be better depends on the data pattern. For example, as we mentioned in the paper, if we meet the Mersenne Twister random distribution, the quantum kernels will be superior.

---

### Official Review · Reviewer_kXVf · 2021-11-05

**Correctness:** 1
**Technical Novelty And Significance:** 1
**Empirical Novelty And Significance:** 2
**Recommendation:** 1
**Confidence:** 5

**Main Review:**

Although the question raised by this paper is interesting, I am not convinced enough by the proposed answer to warrant acceptance. In particular,

**There are many writing flaws**:
- last sentence of 2nd paragraph in unclear
- 1st sentence of 3rd paragraph in unclear: *kernel methods* (and not *the kernel method*) are a family of algorithms
- kernel methods bec*o*me (last word of p.1)
- We *conjecture* (1st contribution bullet point)
- please cite the SVM paper when introducing them (1st par. of sec. 2)
- kernel *trick* (2nd par. sec. 2)
- SVM paragraph: remove *kind of*
- Eq. (4), a $\top$ is missing between the $\phi$ (also support vectors are never defined)
- top of page 4: there is a very clear theoretical answer about what kernel is a valid or not. The good performances of the sigmoid kernel (reference needed here) does not change anything
- I cannot understand what authors mean by *that expands the family of kernel functions*
- Fig.2 seems to imply that tree and graph kernels do not satisfy Mercer's conditions: why such a separation? Also, the ... boxes are not relevant, and "Don't satisfy..." would be a better legend
- the Mersenne Twister distribution could be introduced and commented before the theorem statement
- Eq. (5): I assume $N_l$, $N_m$ instead of $N_1$, $N_2$
- the "number of observations belonging to class X" would be more clear than the *size*
- the $\rightarrow$ vector notation is introduced once and is not consistent with the rest of the paper
- expliciting the formulas in eq. (7) is not necessary

**The exposition of classical and quantum kernels is particularly messy**, which is very harmful as it is a key point in the paper:
- weird $x$ notation
- $f_c$ is mapping $x$ to $\phi(x)$ ?!
- the dot product denoted with both $^T$ and $\cdot$
- note also that to compute a distance, one would use $|| \phi(x) - \phi(y)||$ rather that $<\phi(x), \phi(y)>$ as claimed by the authors
- in the quantum paragraph, I assume $F_q$ means $f_q$
- it seems to me that the definition of $f_q$ from $\mathcal{O}$ to $\mathcal{F}_q$ is repeated. Furthermore, I cannot see any difference from the standard kernel definition
- $f_q$ is mapping $x_i$ to a quantity that depends on $x_j$ ?!
- the $|\cdot >$ notation is very confusing for people used to kernel dot products. Even if it is the standard notation in the quantum literature, why don't we have two $|$ in the definition of $f_q$?
- the circuit $U$ is never defined

In the same vein, **the maths seem not rigorous to me**:
- how can the proof of Theorem 3.1 begin by "we assume" ?! If you have an extra assumption, just put it in the theorem statement
- the same goes for the proof of Theorem 3.2
- the exact statement of Theorem 3.2 should be "there exists $\delta_0$ such that..."
- the CLT does not imply the existence of a $R$ such that the distribution is the standard Gaussian (what if the empirical mean is $1/n$ for instance?)
- what are the "measurements" referring to?
- where do $\sigma$ and $\mu$ play a role?
- please prove "it is easy for a deterministic kernel [...] with small error"

And **the contribution is at best incomplete**, if not erroneous (see comments about the proofs above):
- Theorem 3.1 only states that quantum kernels are superior than standard kernels on this example because the latter cannot learn anything. Can we prove that quantum **can** learn something in this case, i.e., are strictly better. Otherwise the statement is pretty vacuous.
- the definition of $\delta$ is based on empirical quantities, is it suitable? Shouldn't a quantity in expectation make more sense?
- what happens for kernels defined on non-vectorial inputs? Do they have a quantum analog? It should be discussed
- can we use another distance than the Euclidean one?
- if we add an offset to the distance, it seems that the 0.6 threshold exhibited changes, making me dubious about such an absolute value
- Theorem 3.2 actually only shows that quantum kernels are worse than standard ones in a precise regime, not that they are better in the opposite scenario

More generally, the general idea I retain from this paper is that computing quantum kernels requires some randomness, and that if this randomness is greater than the "task randomness/difficulty", it is vain to use quantum kernels. If this message is the right one, I would suggest the authors to be more clear on this, and to drop the $\delta$ criterion, which seems to be a proxy for the task difficulty.

**Summary Of The Paper:**

This paper tries to identify a condition that makes quantum-based kernel methods superior to standard ones.

**Summary Of The Review:**

Although the question raised by this paper is interesting, the proposed answer is not clearly exposed, not rigorous enough on the math side, and not enough significant to warrant acceptance.

---

> ### Author Response · Authors · 2021-11-23
> **Response to reviewer1 (Part)**
>
> Thanks for your time and efforts to review our work. Since the space limitation, we only list several responses here, we upload supplementary material where contains all responses if you are interested.
>
> Q 33: Theorem 3.1 only states that quantum kernels are superior than standard kernels on this example because the latter cannot learn anything. Can we prove that quantum can learn something in this case, i.e., are strictly better. Otherwise the statement is pretty vacuous.
>
> A 33: Thanks for this comment, and we will think about it carefully. Theorem 3.1 shows that quantum kernels are superior to classical kernels when meeting a random distribution based on Mersenne Twister Generator. To show this point, we designed an experiment, and the results are shown in Fig.4(B). The results show that the quantum kernels are almost always better than the classical kernels. As the number of data increases, the quantum kernels will maintain a stable advantage over classical kernels. We think it is an interesting phenomenon. Some researchers already proved that the quantum kernels can be superior to classical kernels when learning a DLP problem. Also, we know learning a Mersenne Twister distribution is a DLP problem. That’s why we assume that the Z-ZZ feature map can effectively simulate the efficacy of the feature map proposed by [10] in the beginning. However, we cannot provide rigorous mathematical proof at present. We will continue to focus and work on this in the future.
>
> Q 34: The definition of δ is based on empirical quantities, is it suitable? Shouldn’t a quantity in expectation make more sense?
>
> A 34: Thank you for this suggestion. In our paper, we try to provide a threshold to decide which one is better to use a quantum kernel method or a classical kernel method. This threshold δ0 is an empirical quantity that is determined through several datasets. It is not an average value or an expectation, but a threshold based on our observation. The ”true” value of δ0 is unknown since it is a value determined by as many datasets as possible. We cannot try all the datasets to make sure of this. But, the existence of δ still makes sense. At least, we know a phenomenon that the variable δ can has some influence to decide whether a quantum kernel method is better or not when compared with a classical kernel method. Based on our experience in 81 datasets, the δ0 will take 0.6.
>
> Q 35: What happens for kernels defined on non-vectorial inputs? Do they have a quantum analog? It should be discussed.
>
> A 35: Thanks for the interesting questions, and we will think about them carefully. To start with, we think it is a very interesting topic to dis- cuss. We list several formats of kernels in Fig 2 though we only discussed the vector kernels in our paper. Take the graph kernel as an example. The idea of graph kernel is to map a graph to some Hilbert space, and the similarity between two graphs can be obtained by the inner product operation in Hilbert space. As far as we can embed a graph into a vector format and use the inner product to represent the similarity, we can apply the quantum kernels methods to it.
>
> Q 37: If we add an offset to the distance, it seems that the 0.6 threshold exhibited changes, making me dubious about such an absolute value.
>
> A 37: We appreciate it for pointing it out. Let me make it clear. We assume what you mean is adding some offset to the distance in the dataset. Yes, the δ of this dataset will be changed because the δ is a value related to distances. However, the threshold δ0 will not be changed. The δ0 is not a value that is determined by a specific dataset, but an empirical value that is determined by several datasets (In our experiment, 81 datasets). For example, assume the δ of a dataset D is 0.55 at first. Based on our theorem, the quantum kernels will be superior to the classical kernels, since the 0.55 < δ0 = 0.6. Then, we add an offset to the distance, and the δ of the dataset D will be changed, for example, 0.7. Since 0.7 > δ0 = 0.6, in the new case using a classical kernel will be a better choice.
>
> Q 38: Theorem 3.2 actually only shows that quantum kernels are worse than standard ones in a precise regime, not that they are better in the opposite scenario.
>
> A 38: Yes, it is. That’s why we make the name ”Deficiencies of the QKM”. We think this work is still worthwhile. Suppose we get the δ of a specific dataset D, if the δ for the D is larger than δ0, we have enough reason to believe that we can use classical kernels to learn this dataset. On the other hand, suppose the δ for the D is less or equal to δ0. Even though we cannot directly say that the quantum kernels will be better, it at least provides us a choice to use quantum kernels. Whether the quantum kernels will be better depends on the data pattern. For example, as we mentioned in the paper, if we meet the Mersenne Twister random distribution, the quantum kernels will be superior.

---

> > ### Comment · Reviewer_kXVf · 2021-11-23
> > **Response**
> >
> > I thank the authors for the answers they provided. Still, the required changes are too important to warrant acceptance. I will stick to my score and encourage the authors to implement the changes they discussed before resubmitting this work to a future venue.

---

### Public Comment · ~Jonas_M._Kübler1 · 2021-11-09
**two related works**

There are two (recent) related works that should be taken into account when analyzing the power of quantum kernels:

1. "Power of data in QML" https://www.nature.com/articles/s41467-021-22539-9
2. "The inductive bias of quantum kernels" https://arxiv.org/abs/2106.03747 (accepted to NeurIPS 2021)

---

### Decision · Program_Chairs · 2022-01-20

**Decision:**

Reject

**Comment:**

There was consensus that though  the paper introduces an interesting question, but not enough exploration has been made. The reviews point out several mathematical in-accuracies, and points out  several issues including that the delta criterion needs to be examined.